# Vision transformer and Mamba-attention fusion for high-precision PCB defect detection

Asim Niaz[1], Muhammad Umraiz[1], Shafiullah Soomro[2], Kwang Nam Choi[1]*

1 Department of Computer Science and Engineering, Chung-Ang University, Seoul, Republic of Korea,
2 Department of Computer Science and Media Technology, Linnaeus University, Vaxjo, Sweden

* knchoi@cau.ac.kr

**Data availability statement:** All relevant data are within the manuscript.

**Funding:** This work was supported by the Ministry of Science and Information and Communication Technology (ICT) and National

## Abstract

Defects in printed circuit boards (PCBs) are being detected using computer vision-based techniques. Defect-free PCBs are essential for the reliability of consumer electronics. However, deep learning-based methods often struggle with imbalanced defect distributions and limited generalization. To address these challenges, we propose ViT-Mamba, a hybrid framework that combines Vision Transformers with a Mamba-inspired attention mechanism for global feature extraction and precise defect segmentation. We further introduce an artificial defect generation module that systematically creates six types of PCB defects to improve robustness. A multiscale hierarchical refinement strategy is employed to enhance feature representation for accurate segmentation. Experiments on a public PCB defect dataset show that ViT-Mamba outperforms existing methods, achieving a mean Average Precision (mAP) of 99.69%.

## Introduction

Defects are unexpected patterns that may emerge across various data modalities, including tabular, visual, textual, and time-series data. They may occur at any time and in any domain, often requiring automated detection for security, access control, or quality assurance. Automated systems are designed to flag such deviations, enabling timely intervention and response [1]. PCB defect detection is vital for maintaining the quality and reliability of consumer electronics, as even minor defects can lead to significant functional failures and economic losses [20]. In 2024, PCBs had an estimated market value of approximately $80 billion. Conventional manual inspection techniques are highly susceptible to environmental factors, leading to decreased defect detection accuracy. Recent advancements in computer vision and deep learning have significantly improved anomaly detection across various domains, including image aesthetics assessment [2], network anomaly detection [3,4], and 3D reconstruction [5]. Transformer-based architectures, such as PIPformers for image inpainting [6] and EAPT for image processing [7], have demonstrated superior feature extraction capabilities, enabling precise identification of complex patterns [8,9]. In medical imaging, models like Ecsu-Net [10] have been effective in segmentation and classification, showcasing the potential of deep learning for intricate defect detection tasks. Similarly, multimodal cascaded CNNs have been

IT Industry Promotion Agency (NIPA) through the High Performance Computing (HPC) Support Project. The funders had no role in study design, data collection and analysis, decision to publish, or preparation of the manuscript.

**Competing interests:** The authors have declared that no competing interests exist.

successfully applied to waste classification [11], highlighting the versatility of convolutional and transformer-based models for anomaly detection.

Detecting small PCB defects is challenging due to their subtle visual characteristics, often leading to visual strain and misclassification. To overcome these issues, researchers have increasingly adopted machine learning techniques, which have significantly advanced PCB defect detection in recent years. Despite the development of various computer vision-based automated methods, accurately identifying defects remains complex and continues to evolve [12,13]. Wang et al. [14] developed an automated algorithm capable of identifying 2 mm pinhole defects within ten seconds. Yuk et al. [15] combined accelerated robust features with a random forest algorithm to enhance defect detection by focusing on feature density in defect-prone areas. Gaidhane et al. [16] employed similarity metrics to detect PCB surface defects, demonstrating effectiveness in identifying and localizing anomalies even in complex component layouts. While other machine learning-based approaches [17,18] have been proposed, they often lack real-time processing capabilities and rely on manual feature extraction, limiting their adaptability to diverse defect types.

Fully supervised detection methods [19–21] are widely used for detecting small targets, leveraging well-established networks such as YOLO [22,23] and TDD-Net [24]. While these approaches achieve high accuracy, their ability to generalize remains limited, making it difficult for the model to recognize PCB defect types not included in the training data. Additionally, training such models demands large-scale annotated datasets, which involve significant labor costs. The imbalanced distribution of defects in real-world PCB datasets further hampers the performance of fully supervised models.

Despite significant advancements in PCB defect detection, existing methods often struggle with limited generalization, computational inefficiency, and imbalanced defect distributions. Fully supervised approaches require extensive labeled datasets, making them costly and labor-intensive, while unsupervised methods frequently lack precision in anomaly localization. Additionally, CNN-based models, though effective in extracting spatial features, often fail to capture global dependencies, which are crucial for complex defect patterns.

To address these challenges, we propose ViT-Mamba, a hybrid framework that combines ViT-based feature extraction with a Mamba-inspired attention mechanism for precise segmentation and anomaly detection. Unlike traditional CNN-based architectures, our method effectively captures long-range dependencies, enhancing its ability to detect subtle and irregular PCB defects. Moreover, our artificial defect creation module systematically augments training data, improving robustness against real-world variations. The multi-scale hierarchical refinement in our decoder further refines feature representations, ensuring accurate anomaly segmentation while maintaining computational efficiency. The key contributions of this work are as follows:

- Artificial Defect Creation Module: Systematically generates diverse PCB defects, mitigating data imbalance issues, and ensuring diverse and realistic training data.
- ViT-Mamba Hybrid Architecture: Combines Vision Transformers for global feature extraction with a Mamba-inspired decoder to improve segmentation precision.
- Multi-Scale Hierarchical Refinement: Utilizes multi-level convolutional features and Mamba attention gates to refine spatial representations, ensuring precise and computationally efficient anomaly segmentation.

The subsequent sections of this paper are organized as follows: the proposed method is first introduced, followed by the experiments and results. Next, the limitations and potential directions for future research are discussed, and finally, the study is concluded.

## Related work

PCB defect detection methods generally fall into two main categories: traditional machine vision techniques and deep learning-based approaches.

Traditional methods typically involve image preprocessing, segmentation, feature extraction, and classification. Moganti et al. [25] surveyed early inspection algorithms that relied on techniques such as image subtraction and template matching. These approaches use hand-crafted features—shape, texture, color—to detect issues like missing holes, open circuits, and shorts. For example, Raihan and Ce [26] applied image subtraction using OpenCV, while Raj and Sajeena [27] utilized general-purpose image processing techniques for defect detection. However, such methods are limited by low adaptability to complex defect types and varying conditions, and they require extensive manual feature engineering [28]

Deep learning has significantly improved defect detection by enabling end-to-end learning from data. Convolutional Neural Networks (CNNs), in particular, have shown strong performance. Chen et al. [28] reviewed CNN-based models like AlexNet, VGG, and ResNet, highlighting their high accuracy. Li and Guo [29] implemented Faster R-CNN with a VGG-16 backbone and data augmentation. Hu and Wang [30] enhanced Faster R-CNN using a Feature Pyramid Network (FPN) and generative adversarial region proposals, improving small defect detection.

Single-stage detectors such as YOLO and SSD offer fast inference, making them suitable for real-time inspection. Xin et al. [31] optimized YOLOv4 with mosaic augmentation and Mish activation. Zhang et al. [32] proposed a lightweight detector using dual attention and a path-aggregation FPN, and Wu et al. [33] introduced GSC YOLOv5 with Ghost Convolution and attention modules to enhance speed and accuracy.

Transformer-based models have recently gained attention for their ability to capture long-range dependencies. An et al. [34] proposed LPViT using label smoothing and patch correlation. Chen et al. [35] replaced CNNs with a Swin Transformer in a YOLO-like architecture. Yang et al. [36] integrated SwinV2TDD into YOLOv7, combining local and global features for improved performance.

Despite progress, challenges remain. Deep models often require large datasets, significant computational resources, and lack interpretability. Traditional methods, while lightweight, fail to generalize across diverse defect types. Recent trends involve hybrid models combining CNNs and Transformers [37], with future directions including GAN-based data augmentation, transfer learning for small datasets, and explainable AI solutions [28].

In summary, the evolution from traditional to deep learning approaches has advanced PCB defect detection. CNNs offer a balance between speed and accuracy, while Transformers improve performance in complex scenarios. Building on this, our proposed method combines mathematically generated artificial defects with a ViT-Mamba architecture. By simulating six realistic defect types, we address data imbalance, and through Vision Transformers and Mamba-inspired attention mechanisms, we enable accurate segmentation and localization—providing a robust, efficient solution for diverse PCB defect detection tasks.

## Proposed method

The proposed method consists of two main modules: artificial defects creation and ViT-Mamba. The overall pipeline of the proposed PCB anomaly detection framework is illustrated in Fig 1.

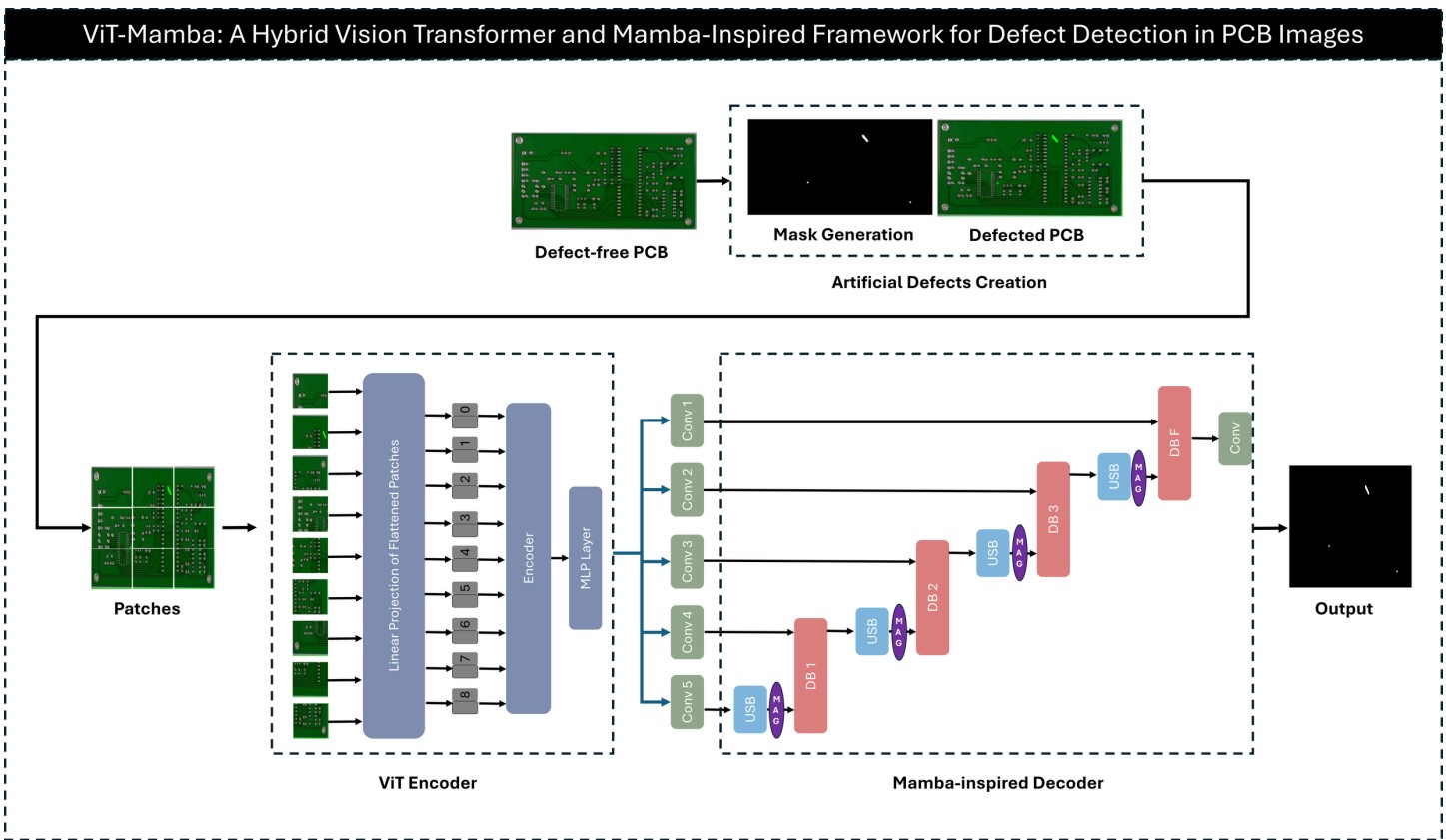

**Fig 1. Pipeline of the ViT-Mamba.** A defect-free PCB undergoes artificial anomaly creation to generate synthetic defects for training. The ViT-Mamba architecture processes these images using a ViT encoder for feature extraction, multi-scale convolutional layers for hierarchical representation, and a Mamba-inspired decoder with attention gates.

## Artificial defects creation

Defect detection in PCBs is inherently challenging due to the limited availability of labeled data for rare defect types and the imbalance in defect categories. To address these challenges, we propose a structured framework for artificially generating six distinct PCB defect types: *Missing Hole*, *Mouse Bite*, *Open Circuit*, *Short*, *Spur*, and *Spurious Copper*. The artificial creation of these defects is governed by mathematically formulated rules to ensure realism, alignment with real-world manufacturing anomalies, and compatibility with diverse PCB designs. Additionally, defects are induced into the dataset using a randomization strategy to enhance variability and robustness.

### 1. Missing hole.

**Condition:** Missing holes occur when predefined vias or mounting holes are not drilled properly during manufacturing. These defects disrupt the electrical connections between PCB layers or impede mechanical assembly.

**Mathematical Formulation:**

$$D_{\text{hole}}(x,y) = \begin{cases} 1 & \text{if } (x,y) \in H, \text{ where } H \text{ is the set of predefined hole centers,} \\ 0 & \text{otherwise.} \end{cases} \quad (1)$$

**Constraint:**

$$(x - h_x)^2 + (y - h_y)^2 < r^2, \quad \text{where } r \text{ is the radius of the hole.} \tag{2}$$

**Explanation:** The predefined hole positions are derived from PCB design files, such as Gerber drill files, which specify the coordinates of critical vias, mounting holes, or component connections. The defect is simulated by applying a circular mask to remove specific hole regions, ensuring the defect aligns with realistic manufacturing errors. Missing hole defects are introduced randomly into selected hole positions.

**2. Mouse bite.**

**Condition:** Mouse bite defects are irregular notches along conductive traces caused by incomplete etching or mechanical stress. These defects can weaken the trace and lead to electrical instability.

**Mathematical Formulation:**

$$D_{\text{mouse}}(x, y) = \begin{cases} 1 & \text{if dist}((x, y), T) < \epsilon \text{ and } R(x, y) = 1, \\ 0 & \text{otherwise.} \end{cases} \tag{3}$$

Here, $T$ is the set of conductive trace points, $\epsilon$ is the max notch distance, and $R(x, y)$ defines the notch shape.

**Explanation:** Mouse bite defects are applied by introducing notches along the conductive traces of the PCB. The trace points $T$ are randomly selected, and the size, shape, and orientation of the notches are varied to simulate real-world imperfections. These defects mimic failures in the etching process and mechanical handling errors.

**3. Open circuit.**

**Condition:** Open circuit defects occur when breaks along conductive traces interrupt the electrical continuity of a circuit, often due to incomplete etching or mechanical damage.

**Mathematical Formulation:**

$$D_{\text{open}}(x, y) = \begin{cases} 0 & \text{if } (x, y) \in T_{\text{break}}, \\ 1 & \text{if } (x, y) \in T \setminus T_{\text{break}}, \\ 0 & \text{otherwise.} \end{cases} \tag{4}$$

Here, $T$ is all trace points, $T_{\text{break}}$ is the broken subset, and $\text{Length}(T_{\text{break}}) < L_{\text{max}}$ limits break length.

**Explanation:** Open circuit defects are simulated by removing segments of conductive traces. The location and length of the break are randomized to replicate diverse failure scenarios. These defects disrupt the continuity of the circuit, providing an effective training dataset for detecting trace interruptions.

**4. Short.**

**Condition:** Short circuit defects are unintended connections between two conductive elements, often caused by excess solder, copper residues, or manufacturing errors.

**Mathematical Formulation:**

$$D_{\text{short}}(x, y) = \begin{cases} 1 & \text{if } (x, y) \in S, \text{ where } S = C_1 \cup C_2, \\ 0 & \text{otherwise.} \end{cases} \tag{5}$$

Here:

- $C_1, C_2$: Sets of points for the two conductive elements being shorted.
- $\text{dist}(C_1, C_2) < \delta$: threshold for short placement.

**Explanation:** Short circuits are introduced by artificially connecting two nearby conductive elements. The placement and geometry of the short are randomized to reflect realistic manufacturing or assembly errors. This defect simulates unintended bridging, a critical anomaly in PCB functionality.

**5. Spur.**

**Condition:** Spur defects are thin, unintended protrusions extending from conductive traces, often caused by over-etching or material deposition errors.

**Mathematical Formulation:**

$$D_{\text{spur}}(x, y) = \begin{cases} 1 & \text{if } (x, y) \in T \text{ and } (x', y') \in P, \\ 0 & \text{otherwise.} \end{cases} \tag{6}$$

Here:

- $T$: Set of trace points.
- $P$: Protrusion points defined as:

$$P = \{(x', y') : \text{dist}((x', y'), T) \leq w \text{ and } \text{Length}(P) \leq l\}. \tag{7}$$

- $w, l$: Maximum width and length of the spur.

**Explanation:** Spur defects are applied as protrusions along the traces, varying in size and shape. The placement of these defects is controlled to ensure alignment with real-world patterns observed in faulty PCBs.

**6. Spurious copper.**

**Condition:** Spurious copper defects occur as random copper patches left on the PCB due to incomplete etching or material deposition.

**Mathematical Formulation:**

$$D_{\text{spurious}}(x, y) = \begin{cases} 1 & \text{if } (x, y) \in C, \\ 0 & \text{otherwise.} \end{cases} \tag{8}$$

Here:

- $C$: Set of randomly placed copper patch points.
- $C = \{(x, y) : R(x, y) = 1\}, where R(x, y)$ is a random binary mask.

**Explanation:** Spurious copper defects are simulated by randomly placing irregular copper patches across the PCB. The randomness ensures variation in patch size, shape, and placement, making the dataset more comprehensive.

**General context constraints.**

1. **Avoid Overlapping Defects:**

$$D_i(x, y) \cap D_j(x, y) = \varnothing \quad \forall i \neq j \tag{9}$$

Ensures that defects do not overlap unless explicitly intended.

2. **Region-Specific Constraints:**

$$D(x, y) \cap \text{EmptyRegion} = \varnothing, \quad D(x, y) \cap \text{CriticalRegion} \neq \varnothing \tag{10}$$

Ensures defects are placed in logical regions of the PCB and avoid unused areas.

**Random induction of defects.** A randomization strategy introduces defects into defect-free PCB images by generating binary masks that define the size, location, and orientation of each defect. The artificially defective PCB image, denoted as $I_a$, is generated by combining the original defect-free image $I$, the defect-only image $M$, and the binary mask $M_a$. The generation formula is defined as follows:

$$I_a = \text{not}(M_a) \odot I + M, \tag{11}$$

where $\text{not}(M_a)$ represents the inverse of the binary mask, marking the non-defective regions, $\odot$ denotes element-wise multiplication. This ensures seamless integration of defects while preserving the intact areas of the PCB.

To further enhance variability, the defects and their corresponding masks undergo a series of random transformations, including rotation, translation, and scaling. Rotation applies a random angle to the defects and masks, translation shifts them randomly along the $x$- and $y$-axes, and scaling adjusts their size using a random factor. This diversifies the dataset, improving model robustness for anomaly detection and segmentation. Figs 2 and 3 illustrate the defect creation pipeline and generated synthetic masks.

## ViT-Mamba

This section presents *ViT-Mamba*, a novel architecture for image anomaly detection and segmentation. The proposed method integrates the global feature extraction capabilities of ViTs in the encoder with a Mamba-inspired decoder that utilizes attention mechanisms for precise segmentation and anomaly localization. The framework is designed to address challenges in industrial and medical imaging, where accurate anomaly detection and pixel-level segmentation are crucial.

**Encoder: Vision transformer-based feature extraction.** The encoder leverages the Vision Transformer (ViT) to capture global dependencies and contextual information by processing images as non-overlapping patches. The input image $\mathbf{X} \in \mathbb{R}^{H \times W \times C}$ is divided into patches of size $P \times P$, forming a sequence of flattened patches $\mathbf{X}_p \in \mathbb{R}^{N \times (P^2 \cdot C)}$, where $N = \frac{H \cdot W}{P^2}$ is the number of patches. The embeddings of these patches are then processed through a multi-layer transformer.

The transformer computes self-attention as follows:

$$\text{Attention}(\mathbf{Q}, \mathbf{K}, \mathbf{V}) = \text{Softmax}\left(\frac{\mathbf{Q}\mathbf{K}^\top}{\sqrt{d_k}}\right)\mathbf{V}, \tag{12}$$

where $\mathbf{Q}$, $\mathbf{K}$, and $\mathbf{V}$ are the query, key, and value matrices, respectively, and $d_k$ is the dimensionality of the keys.

To adapt the ViT for segmentation tasks, the encoded feature maps are passed through lightweight convolutional layers to generate hierarchical feature representations at multiple scales. These feature maps are denoted as $\{b_1, b_2, b_3, b_4, b_5\}$, where $b_1$ corresponds to shallow

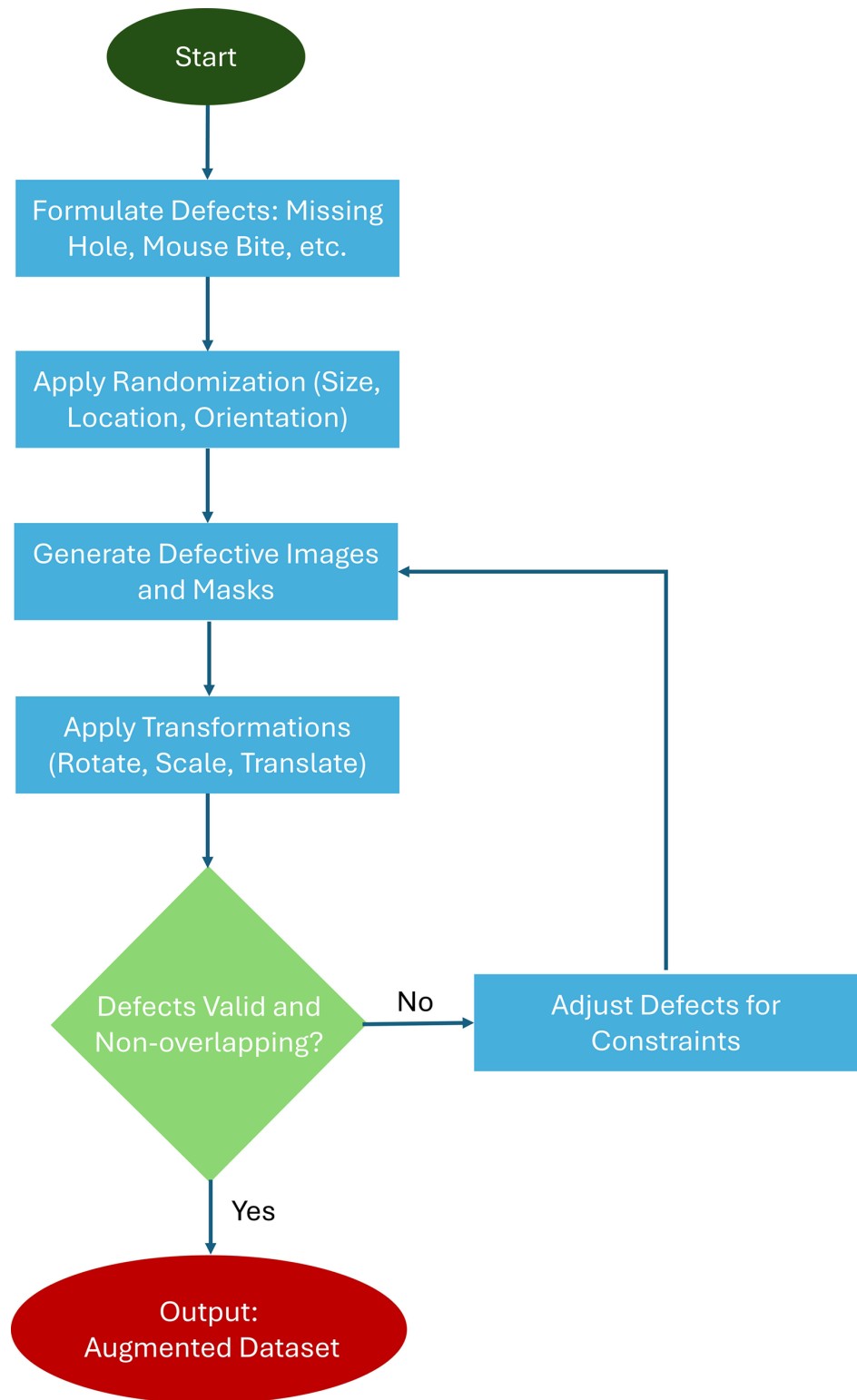

**Fig 2. Artificial defect creation pipeline: A defect-free PCB undergoes mathematically defined defect generation and random transformations to produce defected samples.**

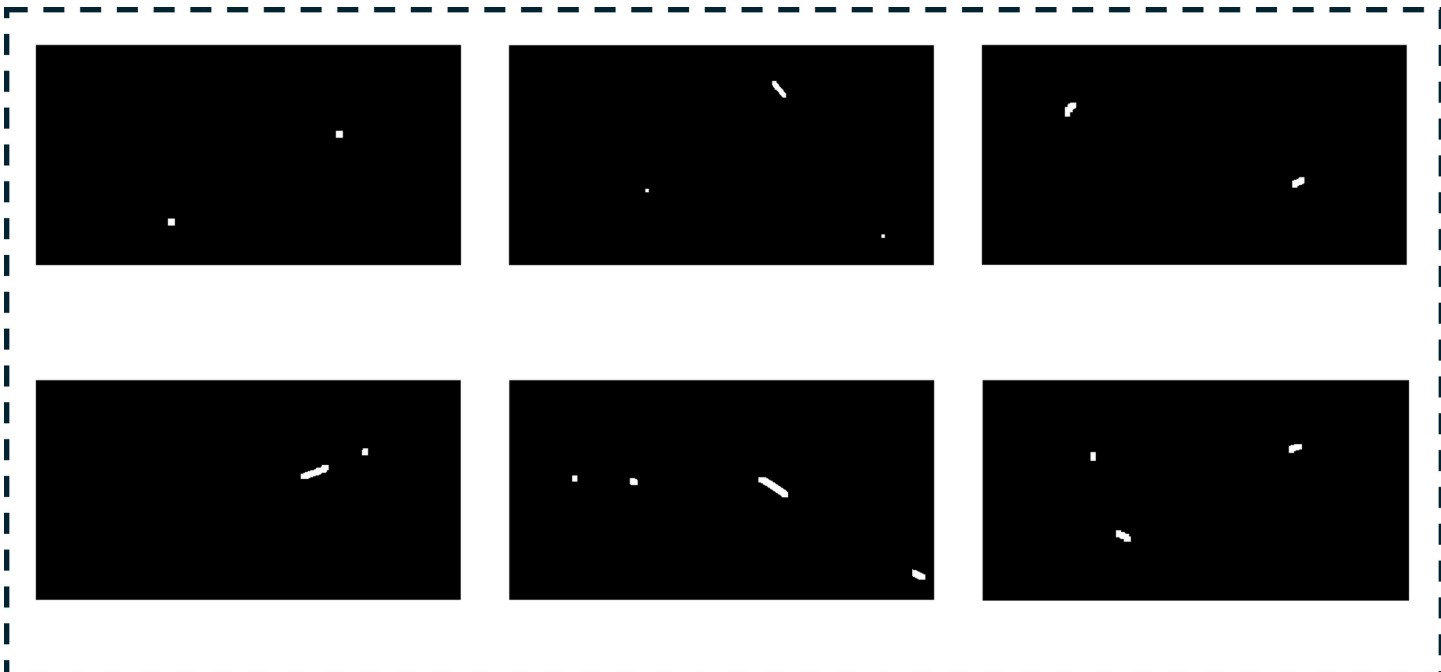

**Fig 3. Illustration of various masks generated using the artificial defects creation module.**

features and $b_5$ contains the most semantic information. The multi-scale features are then forwarded to the decoder.

**Decoder: Mamba-inspired attention-gated design.** The decoder employs a Mamba-inspired design that incorporates attention gates and efficient upsampling mechanisms to produce high-resolution segmentation maps. The decoder comprises three key components:

**Mamba attention gates.** Mamba Attention gates refine the skip connections from the encoder by suppressing irrelevant features and enhancing the most relevant ones. Given the decoder's intermediate features **g** and the encoder's skip connection **x**, the attention gate outputs a refined feature map **x'**:

$$\mathbf{x}' = \mathbf{x} \cdot \psi\left(\text{ReLU}(\mathbf{W}_g\mathbf{g} + \mathbf{W}_x\mathbf{x})\right), \tag{13}$$

where $\mathbf{W}_g$ and $\mathbf{W}_x$ are learnable weight matrices, and $\psi$ is a sigmoid activation function. This mechanism improves the focus on anomalous regions. The Mamba Attention Gate mechanism is illustrated in Fig 4.

**Efficient upsampling.** The decoder progressively upsamples the feature maps using bilinear interpolation followed by convolutional layers. Each upsampling stage integrates refined skip connections from the encoder to recover fine-grained details.

**Multi-scale refinement.** The decoder processes the hierarchical features $b_1, b_2, b_3, b_4, b_5$ starting from the deepest map $b_5$. At each stage, the upsampled feature is concatenated with the corresponding skip connection and passed through a decoding block:

$$\mathbf{f}\text{out} = \text{Conv}\left([\mathbf{f}\text{up}, \mathbf{x}']\right), \tag{14}$$

where $\mathbf{f}_{\text{up}}$ is the upsampled feature, $\mathbf{x}'$ is the refined skip connection, and $[\cdot]$ denotes concatenation.

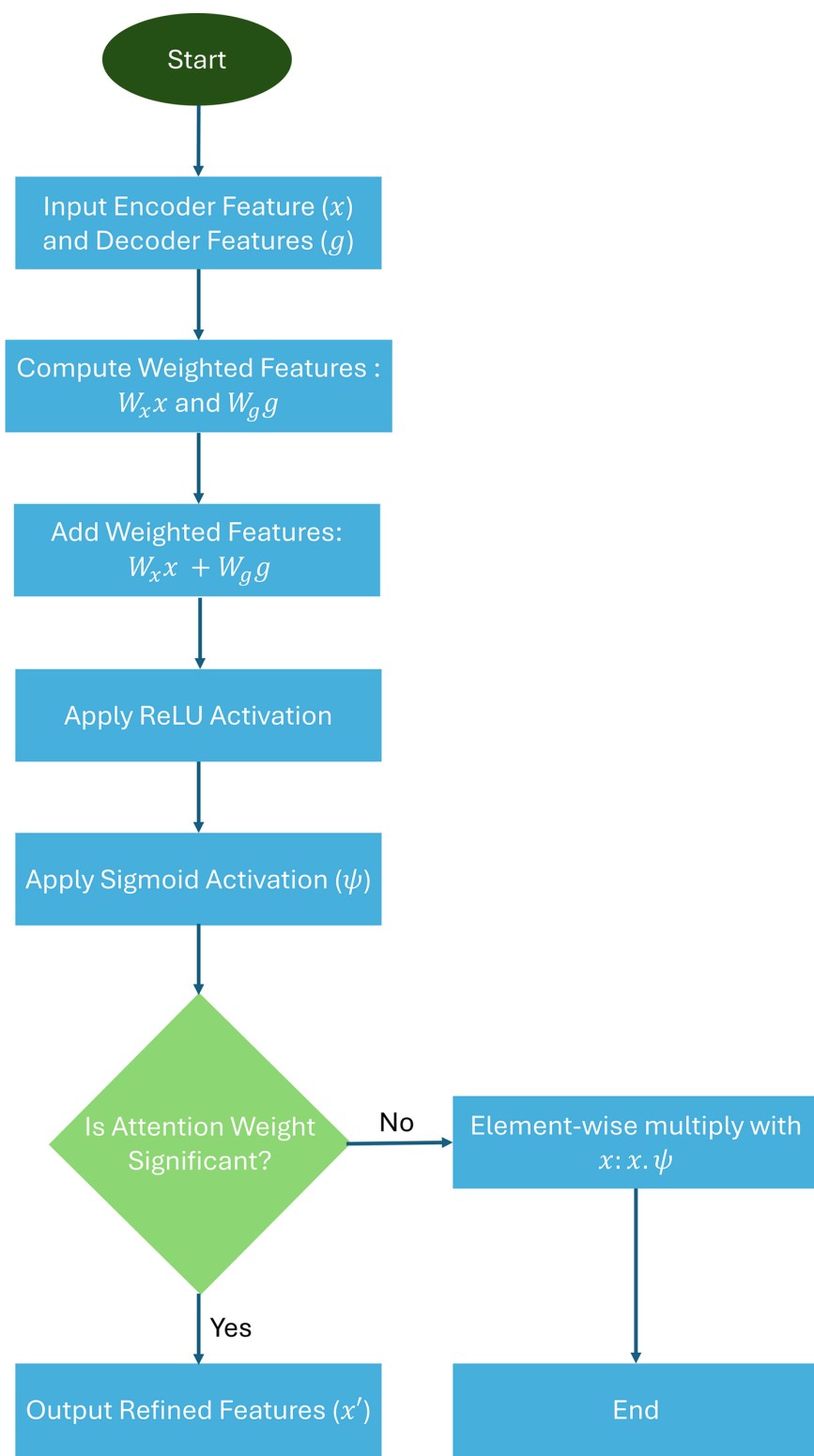

**Fig 4. Mamba attention gate workflow.** Encoder (**x**) and decoder (**g**) features are weighted, combined, and passed through ReLU and sigmoid activations to compute attention weights ($\psi$). These weights refine **x**, producing the final feature map (**x′**) for improved segmentation.

Table 1 summarizes the ViT-Mamba architecture, detailing the encoder, Mamba-based decoder, layer types, output shapes, and key design components for anomaly detection and segmentation.

**Loss calculation.** The proposed ViT-Mamba framework optimizes a hybrid loss function designed to address both segmentation accuracy and anomaly detection performance. The total loss, $\mathcal{L}$, combines pixel-wise segmentation loss, anomaly detection loss, and focal loss to handle challenges such as class imbalance and hard-to-detect anomalies. The loss is defined as:

$$\mathcal{L} = \lambda_1 \mathcal{L}_{\text{seg}} + \lambda_2 \mathcal{L}_{\text{anomaly}} + \lambda_3 \mathcal{L}_{\text{focal}}, \tag{15}$$

where $\lambda_1$, $\lambda_2$, and $\lambda_3$ are weighting factors that balance the contribution of each loss component.

The segmentation loss, $\mathcal{L}_{\text{seg}}$, ensures accurate pixel-wise predictions for both normal and anomalous regions. This loss penalizes the difference between the predicted segmentation mask $\hat{\mathbf{Y}}$ and the ground truth $\mathbf{Y}$. Common choices for $\mathcal{L}_{\text{seg}}$ include the Dice loss or binary cross-entropy (BCE) loss, defined as:

$$\mathcal{L}_{\text{seg}} = -\frac{1}{N} \sum_{i=1}^{N} \left[ y_i \log(\hat{y}_i) + (1 - y_i) \log(1 - \hat{y}_i) \right], \tag{16}$$

where $N$ is the total number of pixels, and $y_i$ and $\hat{y}_i$ represent the ground truth and predicted probabilities, respectively, for pixel $i$.

Table 1. **Network architecture of the ViT-Mamba framework.** The table provides a detailed overview of the components, layer types, output shapes, and additional details.

| Component | Layer Type | Output Shape | Details |
|---|---|---|---|
| Input | Input Image | $256 \times 256 \times 3$ | RGB Input |
| Encoder | ViT Patch Embeddings | $16 \times 16 \times 768$ | Patch Size: $16 \times 16$, Embedding Dim: 768 |
| | Vision Transformer | $16 \times 16 \times 768$ | 12 Layers, 8 Heads, MLP Ratio: 4.0 |
| Convolution | Conv1 | $16 \times 16 \times 128$ | $1 \times 1$ Convolution |
| | Conv2 | $16 \times 16 \times 256$ | $1 \times 1$ Convolution |
| | Conv3 | $16 \times 16 \times 512$ | $1 \times 1$ Convolution |
| | Conv4 | $16 \times 16 \times 1024$ | $1 \times 1$ Convolution |
| | Conv5 | $16 \times 16 \times 1024$ | $1 \times 1$ Convolution |
| Encoder | Skip Connections | – | Feature Maps: $b_1, b_2, b_3, b_4, b_5$ |
| Decoder | Mamba Attention Gate 1 | – | Refines Features: $b_5$ and $b_4$ |
| | Upsampling Block 1 | $16 \times 16 \times 1024 \rightarrow 32 \times 32 \times 1024$ | Bilinear + Conv |
| | Decoding Block 1 | $32 \times 32 \times 1024$ | Conv-BatchNorm-ReLU |
| | Mamba Attention Gate 2 | – | Refines Features: $b_4$ and Decoder Output |
| | Upsampling Block 2 | $32 \times 32 \times 1024 \rightarrow 64 \times 64 \times 512$ | Bilinear + Conv |
| v | Decoding Block 2 | $64 \times 64 \times 512$ | Conv-BatchNorm-ReLU |
| | Mamba Attention Gate 3 | – | Refines Features: $b_3$ and Decoder Output |
| | Upsampling Block 3 | $64 \times 64 \times 512 \rightarrow 128 \times 128 \times 256$ | Bilinear + Conv |
| | Decoding Block 3 | $128 \times 128 \times 256$ | Conv-BatchNorm-ReLU |
| | Mamba Attention Gate 4 | – | Refines Features: $b_2$ and Decoder Output |
| | Final Upsampling | $128 \times 128 \times 256 \rightarrow 256 \times 256 \times 128$ | Bilinear + Conv |
| | Final Decoding Block | $256 \times 256 \times 128$ | Conv-BatchNorm-ReLU |
| | Output Layer | $256 \times 256 \times 3$ | $1 \times 1$ Conv for Final Segmentation |

The anomaly detection loss, $\mathcal{L}_{\text{anomaly}}$, guides the model to distinguish anomalous regions from normal regions. This loss emphasizes the detection of rare and subtle anomalies, penalizing false positives and false negatives. For binary anomaly detection, $\mathcal{L}_{\text{anomaly}}$ can also be formulated using BCE loss.

To address class imbalance and improve the detection of hard-to-classify regions, focal loss, $\mathcal{L}_{\text{focal}}$, is incorporated into the total loss. Focal loss dynamically reduces the weight of well-classified examples, allowing the model to focus on harder examples. It is defined as:

$$\mathcal{L}_{\text{focal}} = -\alpha(1 - p_t)^\gamma \log(p_t), \tag{17}$$

where $p_t$ is the predicted probability for the true class, $\alpha \in [0, 1]$ is a balancing factor, and $\gamma \geq 0$ is the focusing parameter. For multi-class segmentation, this loss can be extended by summing over all classes.

The weighting factors $\lambda_1$, $\lambda_2$, and $\lambda_3$ allow fine-tuning of the relative importance of segmentation accuracy, anomaly detection performance, and hard example emphasis. These weights are determined empirically based on the specific characteristics of the dataset and the task requirements.

The hybrid loss function ensures that the model effectively learns to segment and localize anomalies while addressing challenges like class imbalance and subtle anomaly patterns, making the ViT-Mamba framework robust and efficient for anomaly detection tasks.

The ViT-Mamba framework integrates a Vision Transformer encoder for global context awareness, enhancing robustness to anomalies of varying sizes. Attention gates in the decoder refine skip connections for precise segmentation, while efficient upsampling and multi-scale refinement ensure computational efficiency. Hierarchical features enable accurate anomaly localization, making ViT-Mamba a powerful and scalable solution for image anomaly detection and segmentation. Algorithm 1 outlines the framework's detailed steps.

**Algorithm 1. ViT-Mamba framework for PCB defect detection.**

1: **Input:** Image $I \in \mathbb{R}^{H \times W \times C}$, ViT parameters $\Theta_{ViT}$, Decoder parameters $\Theta_{Dec}$, Loss weights $\lambda_1, \lambda_2, \lambda_3$

2: **Output:** Predicted segmentation mask $\hat{Y}$

3: **1. Preprocessing & Data Augmentation**

4: Introduce artificial defects (e.g., missing hole, spur) and generate binary masks $M_a$

5: **2. Feature Extraction via ViT**

6: Divide $I$ into $P \times P$ patches, obtain embeddings $X_p$, and extract hierarchical features $\{b_1, \dots, b_5\}$ via ViT

7: **3. Decoding via Mamba-Attention**

8: **for** each feature $b_i \in \{b_5, \dots, b_1\}$ **do**

9:   Refine skip connection with attention gates:

10:   $x' = x \cdot \psi(\text{ReLU}(W_g g + W_x x))$

11:   Upsample, concatenate, and pass through decoding blocks

12: **end for**

13: **4. Segmentation Output**

14: Apply $1 \times 1$ convolution to obtain $\hat{Y}$

15: **5. Loss Calculation**

16: Compute loss: $L = \lambda_1 L_{seg} + \lambda_2 L_{anomaly} + \lambda_3 L_{focal}$

17: **6. Optimization**

18: Update $\Theta_{ViT}, \Theta_{Dec}$ via backpropagation

       **return** Predicted segmentation mask $\hat{Y}$

## Experiments and results

### Dataset and implementation details

We utilize the publicly available PCB defect dataset[38] released by the Open Lab on Human-Robot Interaction at Peking University. This dataset comprises 1,386 images containing six types of PCB defects, namely missing hole, mouse bite, open circuit, short, spur, and spurious copper. These defects are introduced into PCB images, making the dataset suitable for tasks such as defect detection, classification, and registration. Fig 5 presents examples from the PCB defect dataset, illustrating one image per defect category with annotated bounding boxes.

During the experimental evaluation, the model was trained for 1000 epochs. The batch size was set to 8, and the learning rate was initialized at 0.0001. To address potential overfitting due to the limited size of the training dataset, data augmentation was employed by applying random rotations within the range of −45 to 45 degrees. The proposed network was implemented using PyTorch, and all experiments were conducted on a workstation equipped with an NVIDIA GeForce RTX 3090Ti GPU.

### Comparison with state-of-the-art methods

Table 2 presents a comparative analysis of various supervised and unsupervised methods for PCB defect detection, evaluated across six defect types: Missing Hole, Mouse Bite, Open Circuit, Short, Spur, and Spurious Copper. The results are reported in terms of Average Precision (AP) for each defect type, along with the mean Average Precision (mAP) for overall performance comparison.

Among the SVM-based supervised approaches, Chaudhary et al. achieved the highest mAP of 91.7%, significantly outperforming Zhang et al. and Li et al., whose mAP scores were

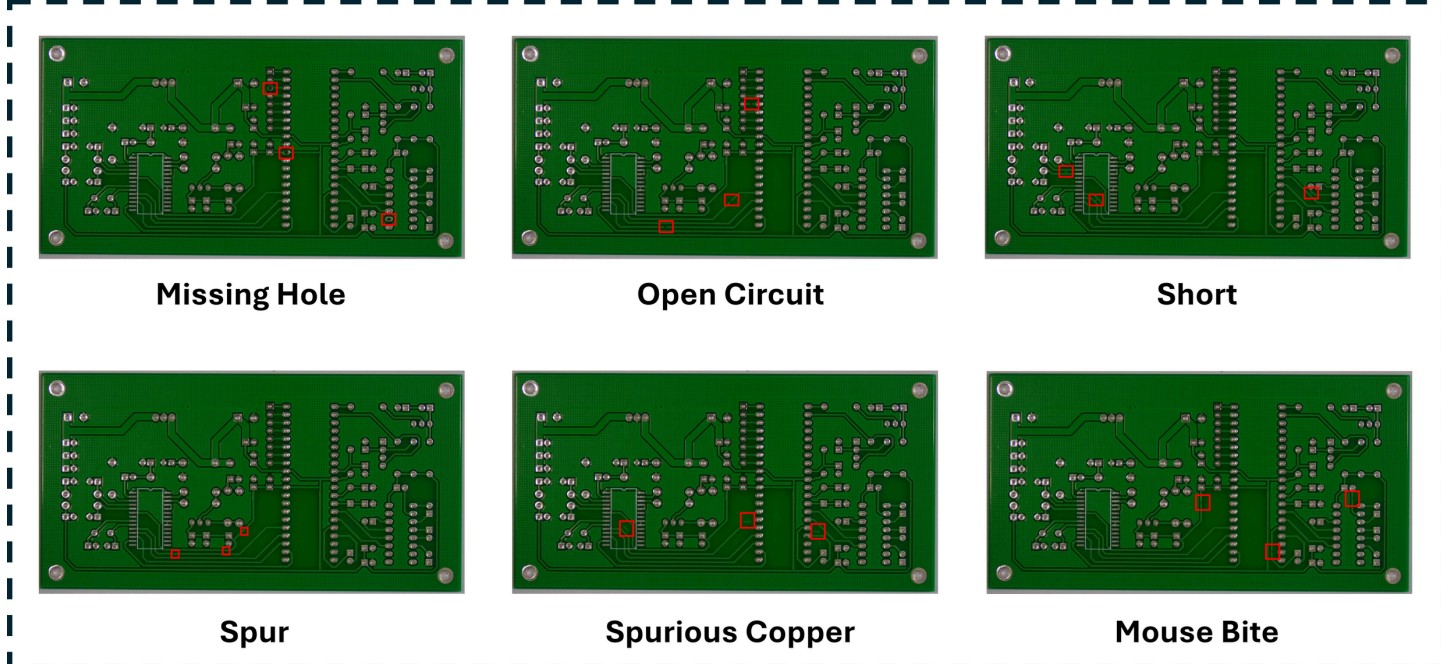

**Fig 5. Examples from the PCB defect dataset, showcasing one image per defect category with annotated bounding boxes: missing hole, mouse bite, open circuit, short, spur, and spurious copper.**

**Table 2. Comparison of different methods for PCB defect detection.**

| Method | AP | | | | | | mAP |
|---|---|---|---|---|---|---|---|
| | Missing Hole | Mouse Bite | Open Circuit | Short | Spur | Spurious Copper | |
| **SVM (Supervised)** | | | | | | | |
| Zhang et al. [39] | 44.4 | 27.5 | 56.9 | 82.2 | 44.2 | 82.1 | 56.2 |
| Li et al. [40] | 46.6 | 13.0 | 36.2 | 48.6 | 31.2 | 65.7 | 40.4 |
| Chaudhary et al. [41] | 97.2 | 84.7 | 96.0 | 92.6 | 82.7 | 97.2 | 91.7 |
| **Neural Network (Supervised)** | | | | | | | |
| Li et al. [42] | 97.0 | 97.9 | 97.0 | 97.5 | 93.7 | 98.5 | 97.7 |
| Huang et al. [43] | 97.0 | 97.9 | 97.0 | 97.5 | 93.7 | 98.5 | 97.5 |
| Lim et al. [44] | - | - | - | - | - | - | 99.17 |
| Hu et al. [45] | - | - | - | - | - | - | 98.45 |
| Ding et al. [24] | - | - | - | - | - | - | 98.9 |
| Chen et al. [46] | - | - | - | - | - | - | 99.17 |
| **Unsupervised** | | | | | | | |
| Zeng et al. [47] | 97.35 | 97.15 | 95.46 | 99.15 | 99.82 | 99.05 | 93.27 |
| Roth et al. [48] | 85.14 | 60.06 | 80.18 | 85.32 | 50.94 | 81.09 | 73.77 |
| Changlin et al. [49] | 99.91 | 97.55 | 98.42 | 99.97 | 99.92 | 99.97 | 99.29 |
| ViT Mamba | 99.93 | 99.21 | 99.13 | 99.96 | 99.98 | 99.97 | 99.69 |

56.2% and 40.4%, respectively. The Neural Network-based supervised approaches demonstrated higher accuracy, with Li et al. and Huang et al. both achieving an mAP of 97.7% and 97.5%, respectively. Other deep learning-based methods, such as Lim et al. and Chen et al., reported mAP values exceeding 99.0%, indicating strong performance in supervised learning settings.

In contrast, unsupervised methods varied significantly in performance. While Roth et al. achieved a relatively lower mAP of 73.77%, other methods such as Zeng et al. (93.27%) and Changlin et al. (99.29%) showed strong results. The proposed method, ViT-Mamba, outperformed all existing techniques, achieving the highest mAP of 99.69%, with near-perfect AP values across all defect categories. These results highlight the effectiveness of ViT-Mamba in accurately detecting PCB defects without requiring extensive supervised training. The radar chart in Fig 6 illustrates the AP scores of four top-performing PCB defect detection methods (Changlin et al., ViT Mamba, Zeng et al., and Chaudhary et al.) across six defect categories. Each axis represents a different defect type, and the area covered by each method indicates its overall detection capability. ViT Mamba and Changlin et al. exhibit consistently high performance across all defect types, while Zeng et al. shows slightly lower accuracy in detecting open circuits. The chart highlights variations in method effectiveness, offering an intuitive comparison of strengths and weaknesses in PCB defect detection.

The significant improvement in performance demonstrates the potential of ViT-Mamba as a robust and scalable solution for real-world PCB defect detection. The results indicate that transformer-based architectures, particularly ViT-Mamba, can generalize well to diverse defect types while maintaining superior detection accuracy.

## Ablation studies

To assess the contribution of each component in the proposed ViT-Mamba framework, we conducted ablation experiments, as per Table 3, on the PCB defect dataset. The study isolates the impact of three core modules: (1) the Vision Transformer (ViT) encoder, (2) Mamba-inspired attention gates, and (3) the multiscale hierarchical refinement strategy. In each

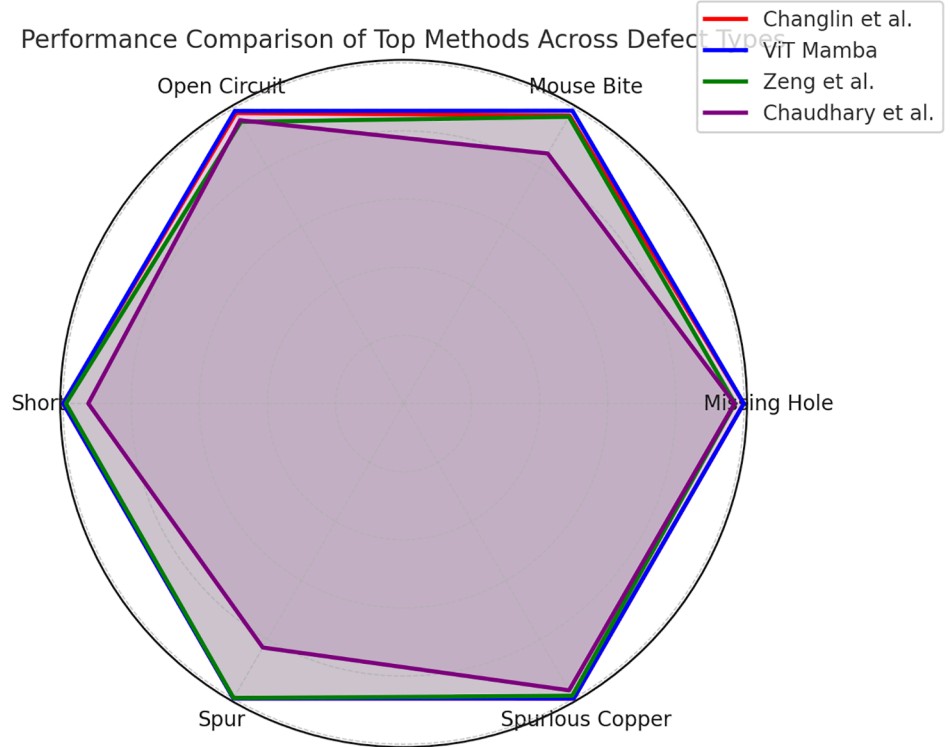

**Fig 6. Performance comparison of top methods across defect types.**

**Table 3**. **Performance comparison from ablation experiments.** Each component is removed or replaced to assess its contribution to the model. Results are reported as mean Average Precision (mAP).

| Configuration | mAP (%) |
|---|---|
| Full ViT-Mamba (Proposed) | **99.69** |
| ViT Replaced with ResNet Backbone | 90.58 |
| Without Mamba Attention Gates (simple skip) | 96.47 |
| Without Multiscale Refinement (single scale) | 95.83 |

experiment, one component was removed or replaced, while the rest of the model remained unchanged.

The full ViT-Mamba model achieved the highest accuracy. Removing the artificial defect generation module significantly degraded performance, emphasizing its role in addressing data imbalance. Replacing Mamba attention with simple skip connections reduced segmentation precision, and removing multiscale refinement affected the model's ability to recover fine-grained details. Substituting the ViT encoder with a ResNet backbone also resulted in lower performance, indicating the importance of modeling global dependencies.

These results confirm the effectiveness of each proposed module and validate the overall architecture design of ViT-Mamba for high-precision PCB defect detection.

## Discussion

The ViT-Mamba framework, while highly accurate, could face challenges in computational efficiency and real-time deployment, especially on edge devices. Its reliance on ViTs

increases processing demands compared to CNNs, and its generalization to real-world PCB images with diverse lighting, resolutions, and textures remains untested. Industrial variations like noise, reflections, and occlusions could impact robustness, requiring further validation. The artificial defect creation module, though useful, may not fully capture all real-world defect patterns, affecting adaptability.

Future research should focus on optimizing computational efficiency through lightweight transformers, model compression (pruning, quantization), and edge AI implementations. Unsupervised and self-supervised learning can enhance domain generalization, while few-shot and zero-shot learning could improve adaptability to novel defects. Multi-modal imaging (thermal, X-ray) may further refine defect detection, and adversarial training can boost robustness against industrial variations. Lastly, explainable AI (XAI) will improve model transparency and trust, supporting broader adoption in automated PCB quality inspection for consumer electronics manufacturing.

## Conclusion

This study proposed ViT-Mamba, a hybrid Vision Transformer (ViT) and Mamba-inspired framework for PCB defect detection. By integrating global feature extraction with attention-driven segmentation, the model effectively detects subtle and irregular defects while maintaining computational efficiency. The inclusion of an Artificial Defect Creation Module further enhances robustness by diversifying training data. Experimental results show that ViT-Mamba outperforms existing methods, achieving a mean Average Precision (mAP) of 99.69%, demonstrating its effectiveness for real-world PCB inspection. Future work can focus on optimizing computational efficiency and enhancing generalization to unseen defects. ViT-Mamba contributes to advancing deep learning for industrial anomaly detection, offering a scalable and accurate solution for PCB quality inspection.

## Author contributions

**Conceptualization:** Asim Niaz, Muhammad Umraiz.

**Data curation:** Asim Niaz, Muhammad Umraiz.

**Formal analysis:** Asim Niaz, Muhammad Umraiz.

**Funding acquisition:** Kwang Nam Choi.

**Investigation:** Asim Niaz.

**Methodology:** Asim Niaz, Muhammad Umraiz.

**Resources:** Kwang Nam Choi.

**Software:** Asim Niaz, Muhammad Umraiz.

**Supervision:** Kwang Nam Choi.

**Validation:** Asim Niaz, Muhammad Umraiz.

**Visualization:** Asim Niaz, Muhammad Umraiz.

**Writing – original draft:** Asim Niaz, Muhammad Umraiz, Kwang Nam Choi.

**Writing – review & editing:** Asim Niaz, Shafiullah Soomro.

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
