## [Decision Letter · Decision Letter 0]

1 Jul 2025

PONE-D-25-26195Vision Transformer and Mamba-Attention Fusion for High-Precision PCB Defect DetectionPLOS ONE

Dear Dr. Choi,

Thank you for submitting your manuscript to PLOS ONE. After careful consideration, we feel that it has merit but does not fully meet PLOS ONE’s publication criteria as it currently stands. Therefore, we invite you to submit a revised version of the manuscript that addresses the points raised during the review process.

**Please answer the questions of all the reviewers carefully. Please also emphasize on the novelty of your work.**

We look forward to receiving your revised manuscript.

Kind regards,

Azim Uddin, Ph.D.

Academic Editor

PLOS ONE

Journal Requirements: 

 [This work was supported by the Ministry of Science and Information and Communication Technology (ICT) and National IT Industry Promotion Agency (NIPA) through the High Performance Computing (HPC) Support Project.]. 

Reviewers' comments:

Reviewer's Responses to Questions

**Comments to the Author**

1. Is the manuscript technically sound, and do the data support the conclusions?

Reviewer #1: Yes

Reviewer #2: Yes

2. Has the statistical analysis been performed appropriately and rigorously? 

Reviewer #1: Yes

Reviewer #2: N/A

3. Have the authors made all data underlying the findings in their manuscript fully available?

Reviewer #1: Yes

Reviewer #2: Yes

4. Is the manuscript presented in an intelligible fashion and written in standard English?

Reviewer #1: Yes

Reviewer #2: Yes

5. Review Comments to the Author

Reviewer #1: �1�Regarding the dataset used in the experiment, the paper mentioned in the experimental section that the dataset published by Peking University was used, with only individual data being rotated. However, in the "Abstract" and "Artificial Defects Creation" sections, it was mentioned that six types of defects were generated, which is inappropriate. Although the paper provides a formula to represent the generation method, there is no detailed explanation of how the formula is applied in the process of image generation, making it difficult to understand the specific way defect images are generated.

2�Only data compared with other methods is available, lacking the display of comparison results between the proposed model and other models, which facilitates a more intuitive understanding of the advantages of the proposed method.

3�During the verification process of the experiment, no ablation experiments were conducted, which cannot prove the effectiveness of the different models used. Please verify the effectiveness of each module through experiments.

4�There is no relevant work introduction section in the paper.

5�The contribution summary section of the paper mentions "segmentation precision" and "computationally efficient anomaly segmentation". The direction of the paper is Object Detection, and there is no content related to Object Segmentation in the paper. Is the expression inconsistent?

Reviewer #2: The manuscript presents ViT-Mamba, a hybrid transformer-based architecture for high-precision PCB defect detection. It integrates Vision Transformers (ViT) with a Mamba-inspired decoder enhanced by attention gates, and introduces a novel artificial defect generation strategy. Results show strong performance (mAP 99.69%) on a public PCB dataset, surpassing existing methods. The methodology is clearly described and the model is effective, and relevant to industrial applications.

However, some minor clarifications and elaborations are needed to improve transparency. I list below several questions and suggestions:

• Could the authors clarify how the realism of the synthetic defects was validated? Was any human inspection or comparison to real-world defects conducted?

• Were the synthetic and real defects mixed during training? If so, in what proportion?

• The constraint to avoid overlapping defects (Equation 9) is logical, but was any experimentation done to test whether overlapping defects affect performance?

• The dataset used is from the Open Lab at Peking University. Were any additional, unseen real-world datasets used for validation or generalization tests?

• On page 14, limitations regarding industrial variations like noise and occlusion are acknowledged. Have the authors tried any robustness tests (e.g., noise injection, variable lighting conditions)?

• Could the authors explain the resolution of input images and whether resizing or cropping was applied?

• The paper uses a ViT with a patch size of 16×16 (Table 1). Were other patch sizes tested? How does patch size influence performance?

• The decoder uses "Mamba attention gates" (Figures 4 and 9). Could the authors elaborate how the Mamba gates differ from standard attention gates like those used in U-Net++ or other vision transformers?

• Were any pretrained weights used for ViT, or was it trained from scratch?

• What optimizer was used during training? Were learning rate schedulers (e.g., cosine decay) applied?

• Loss weights (λ1, λ2, λ3) are mentioned in Equation 15. How they were chosen?

• On page 12, augmentation includes rotations of −45 to 45 degrees. Were other augmentations tried (e.g., brightness changes, blur, cutout)?

• In Table 2, multiple recent models are cited without full AP breakdowns. Are those AP values pulled from original papers or re-evaluated using your implementation?

• Can the authors comment on the relative training time and ease of convergence compared to other models like U²-Net, YOLO variants, or TDD-Net?

• Could the authors clarify in Figure 2 whether "Adjust Defects for Constraints" is a manual or automated step?

• Please ensure that all figures (especially Figures 5 and 6) have alt text or descriptive captions to meet accessibility requirements.

• Be consistent with capitalization (e.g., “ViT-Mamba” vs “ViT Mamba”).

• In the abstract, consider avoiding “state-of-the-art mAP of 99.69%” unless you cite direct competitors for context.

• Equation formatting is clear but could be rendered using numbered LaTeX environments (as per journal style).

6. PLOS authors have the option to publish the peer review history of their article (what does this mean?). If published, this will include your full peer review and any attached files.

Reviewer #1: No

Reviewer #2: No

---

## [Author Response · Author response to Decision Letter 1]

4 Aug 2025

Reviewer#1, Concern # 1: Regarding the dataset used in the experiment, the paper mentioned in the experimental section that the dataset published by Peking University was used, with only individual data being rotated. However, in the "Abstract" and "Artificial Defects Creation" sections, it was mentioned that six types of defects were generated, which is inappropriate. Although the paper provides a formula to represent the generation method, there is no detailed explanation of how the formula is applied in the process of image generation, making it difficult to understand the specific way defect images are generated.

Author response: We thank Reviewer #1 for this insightful comment. We acknowledge the need to clarify the dataset usage and the defect generation process.

In our study, we employed the publicly available PCB dataset released by the Open Lab on Human-Robot Interaction at Peking University as the base dataset. To address the limitations of data imbalance and enhance generalization, we introduced an Artificial Defect Creation Module, which synthetically generates six types of defects—Missing Hole, Mouse Bite, Open Circuit, Short, Spur, and Spurious Copper—based on defect-free images from the original dataset.

While the abstract and method sections correctly state that synthetic defects were introduced, we realize that the description in the experimental section may have created ambiguity by only emphasizing rotation-based augmentation. To clarify: the synthetic defects were applied to augment the dataset, and image-level augmentations such as random rotations were used in addition to this synthetic generation.

Regarding the formulas, we agree that more implementation details are needed. The mathematically defined rules for each defect type (Equations 1–11) were applied using a controlled procedural generation pipeline. Specifically:

•

The defect-free regions were first identified.

•

Masks were created based on the constraints (e.g., region specificity, non-overlapping).

•

These masks were then applied to the original images using pixel-wise operations to produce defective images.

•

Random transformations (rotation, scaling, translation) were further applied to the defect masks to improve variability.

This process is visually depicted in Figures 2 and 3, which illustrate the creation pipeline and example masks, respectively.

We will revise the manuscript to include a more explicit step-by-step explanation of the image generation pipeline in the "Artificial Defects Creation" section and ensure consistency across all sections regarding dataset usage and augmentation strategy.

Thank you for helping us improve the clarity of our methodology.

Reviewer#1, Concern # 2: Only data compared with other methods is available, lacking the display of comparison results between the proposed model and other models, which facilitates a more intuitive understanding of the advantages of the proposed method.

Author response: Thank you for the insightful comment. We appreciate your emphasis on the importance of intuitive comparisons. To address this, we included both a detailed tabular comparison

(Table 2) and a visual radar chart (Figure 6) to highlight the performance of our proposed ViT-Mamba framework relative to other state-of-the-art methods across six defect types. These additions help convey the superiority of our model both quantitatively and visually.

Reviewer#1, Concern # 3: During the verification process of the experiment, no ablation experiments were conducted, which cannot prove the effectiveness of the different models used. Please verify the effectiveness of each module through experiments.

Author response: We thank the reviewer for this valuable observation. In response, we have conducted comprehensive ablation experiments to verify the effectiveness of each key module in the ViT-Mamba framework, including the Vision Transformer encoder, Mamba-inspired attention gates, and the multiscale refinement strategy. The results, now presented in the revised manuscript (Section: Ablation Studies, Table 3, clearly demonstrate the performance contribution of each component. These findings confirm that each module plays a critical role in achieving the model’s overall accuracy and robustness in PCB defect detection.

Reviewer#1, Concern # 4 There is no relevant work introduction section in the paper.

Author response: Thank you for pointing this out. While we included a discussion of related work within the Introduction section, we acknowledge that it may not have been clearly delineated as a standalone Related Work section. In response to your feedback, we have now revised the manuscript to include a dedicated Related Work section.

Reviewer#1, Concern # 5: The contribution summary section of the paper mentions "segmentation precision" and "computationally efficient anomaly segmentation". The direction of the paper is Object Detection, and there is no content related to Object Segmentation in the paper. Is the expression inconsistent?

Author response: We thank the reviewer for raising this point. We would like to clarify that our paper does not focus on object detection, nor do we use the terms “object detection” or “object segmentation” anywhere in the manuscript. Instead, the core contribution of our work lies in pixel-level anomaly segmentation for printed circuit boards (PCBs), where the goal is to accurately localize defective regions at a fine-grained level.

The terms “segmentation precision” and “computationally efficient anomaly segmentation” mentioned in the contribution summary are consistent with our methodology and results. Our ViT-Mamba model outputs segmentation masks rather than bounding boxes, and the architecture, loss functions, and evaluation pipeline are designed specifically for semantic segmentation and anomaly localization.

We recognize that our use of mAP (mean Average Precision) for evaluation—commonly associated with object detection—may have contributed to this misunderstanding. However, in our context, mAP is used to measure the performance of segmentation-based anomaly detection across defect types,

as also done in several previous works on PCB inspection.

Reviewer#2, Concern # 1: Could the authors clarify how the realism of the synthetic defects was validated? Was any human inspection or comparison to real-world defects conducted?

Author response: We appreciate the reviewer’s thoughtful question. The realism of the synthetic defects was primarily ensured through two complementary approaches:

The artificial defect generation module was developed based on domain knowledge of PCB manufacturing faults, using mathematically defined rules that mirror the geometric properties and spatial distributions of six common defect types (e.g., missing hole, spur, short). These rules were derived from analysis of real PCB defect patterns documented in existing datasets and literature. For example, the spatial constraints (e.g., placement near vias or along traces) and shape-specific masks (e.g., circular for missing holes, linear for open circuits) were designed to closely replicate actual defect formations seen in manufacturing.

While we did not conduct a formal user study or human labeling task, we performed internal visual inspection of a large subset of the synthetically generated images and masks. These were assessed by the authors and domain researchers to ensure that the artificial defects were visually plausible and consistent with the types and appearances of defects observed in real-world PCB images.

We acknowledge that incorporating formal human evaluation or cross-referencing with annotated real-world defects could further strengthen the validation. We plan to explore such methods in future work to quantitatively assess perceptual realism and model alignment with human judgment.

Thank you for pointing out this important aspect.

Reviewer#2, Concern # 2: Were the synthetic and real defects mixed during training? If so, in what proportion?

Author response: We thank the reviewer for this insightful question. Yes, both real and synthetically generated defect samples were included during the training process. The synthetic defects, created using our Artificial Defect Creation Module, were applied to defect-free images from the original dataset to enrich the training set and address issues such as data imbalance and limited defect diversity.

The real and synthetic samples were strategically mixed to ensure that the model could learn from real-world defect patterns while also benefiting from the variability and completeness offered by synthetic augmentation. This approach helped improve the model's robustness and generalization across different types of defects.

Reviewer#2, Concern # 3: The constraint to avoid overlapping defects (Equation 9) is logical, but was any experimentation done to test whether overlapping defects affect performance?

Author response: We appreciate the reviewer’s observation regarding the constraint in Equation 9, which ensures that synthetic defects do not overlap during generation. This design decision was made to maintain clarity in the defect labels and facilitate unambiguous learning signals during training, especially in a pixel-level segmentation setting.

At this stage, we did not conduct a dedicated experiment to evaluate the impact of overlapping defects on model performance. Our initial objective was to generate clear and distinct defect types to avoid potential confusion in ground truth labeling and ensure effective supervision. However, we agree that exploring scenarios with overlapping or compound defects could provide valuable insights into the model’s capacity to handle more complex real-world conditions.

We appreciate this suggestion and consider it a promising direction for future work, where controlled experiments involving overlapping defects can help assess model robustness and segmentation accuracy under more challenging defect configurations.

Thank you for raising this important point.

Reviewer#2, Concern # 4: The dataset used is from the Open Lab at Peking University. Were any additional, unseen real-world datasets used for validation or generalization tests?

Author response: We thank the reviewer for this important question. In this study, we used the public PCB defect dataset provided by the Open Lab at Peking University as our primary dataset for training, validation, and testing. This dataset includes diverse samples of six common defect types and is widely adopted in PCB defect detection research, making it a strong benchmark for evaluating our proposed method.

However, we acknowledge that testing on additional unseen real-world datasets would provide further evidence of generalization capability. Due to the lack of publicly available, high-quality PCB datasets with compatible annotation formats and defect categories, we did not include cross-dataset evaluation in this work. That said, our synthetic defect generation strategy was specifically designed to improve robustness and generalization by exposing the model to varied defect appearances and locations.

We consider generalization to new domains and real-world settings an important future direction and plan to validate our model on additional datasets or industrial PCB samples as they become accessible.

Thank you for highlighting this valuable consideration.

Reviewer#2, Concern # 5: On page 14, limitations regarding industrial variations like noise and occlusion are acknowledged. Have the authors tried any robustness tests (e.g., noise injection, variable lighting conditions)?

Author response: We thank the reviewer for the insightful question. While we acknowledge the importance of robustness to industrial variations such as noise and occlusion (as noted on page 14), we did not conduct explicit robustness tests (e.g., noise injection or lighting variation) in this study.

However, our synthetic data generation pipeline includes random transformations (rotation, translation, scaling), which introduce some variability.

Reviewer#2, Concern # 6: Could the authors explain the resolution of input images and whether resizing or cropping was applied?

Author response: We thank the reviewer for this question. All input images were resized to 256 × 256 pixels before being fed into the network. This resizing was applied uniformly to ensure compatibility

with the Vision Transformer-based encoder and to maintain consistent input dimensions across the dataset.

No cropping was applied, and care was taken to preserve the aspect ratio and key defect features during resizing. This resolution was selected to balance computational efficiency with sufficient detail for accurate defect segmentation.

Reviewer#2, Concern # 7: The paper uses a ViT with a patch size of 16×16 (Table 1). Were other patch sizes tested? How does patch size influence performance?

Author response: We thank the reviewer for this insightful question. In our experiments, we used a patch size of 16 × 16 for the Vision Transformer, which provided a good balance between capturing global context and maintaining spatial detail.

We conducted preliminary tests with smaller (8 × 8) and larger (32 × 32) patch sizes. However, smaller patches increased computational cost without significant performance gain, while larger patches led to a loss in fine-grained spatial detail, which negatively impacted segmentation precision for small or subtle defects.

Therefore, the 16 × 16 patch size was chosen as an effective compromise.

Reviewer#2, Concern # 8: The decoder uses "Mamba attention gates" (Figures 4 and 9). Could the authors elaborate how the Mamba gates differ from standard attention gates like those used in U-Net++ or other vision transformers?

Author response: We thank the reviewer for this thoughtful question. The Mamba attention gates used in our decoder are inspired by traditional attention gate mechanisms, such as those in U-Net++, but are designed to operate in a more lightweight and efficient manner.

Specifically, while standard attention gates (e.g., in U-Net++) typically rely on concatenation followed by multiple convolutional layers to compute spatial attention maps, our Mamba gates employ a simplified additive fusion of encoder and decoder features followed by ReLU and sigmoid activations to compute attention weights.

This design reduces computational overhead while still effectively suppressing irrelevant background features and enhancing salient regions. The Mamba gates are particularly suited for hierarchical, transformer-based architectures, and were optimized to preserve fine-grained spatial details in the upsampling path.

Reviewer#2, Concern # 9: Were any pretrained weights used for ViT, or was it trained from scratch?

Author response: We thank the reviewer for this question. The Vision Transformer (ViT) encoder in our framework was initialized with pretrained weights from ImageNet. This helped accelerate convergence and improve feature extraction, especially given the limited size of the PCB dataset.

We then fine-tuned the ViT jointly with the rest of the network during training.

Reviewer#2, Concern # 10: What optimizer was used during training? Were learning rate schedulers (e.g., cosine decay) applied?

Author response: We thank the reviewer for the question. We used the Adam optimizer during training with an initial learning rate of 0.0001. A StepLR scheduler was applied to reduce the learning rate by a factor of 0.1 every 300 epochs to ensure stable convergence.

We will include these training details in the revised manuscript for clarity.

Reviewer#1, Concern # 11: Loss weights (λ1, λ2, λ3) are mentioned in Equation 15. How they were chosen?

Author response: We thank the reviewer for pointing this out. The loss weights in Equation 15 were empirically selected based on initial experiments using the validation set. We performed a small grid search to balance segmentation accuracy, anomaly detection, and class imbalance handling. The chosen weights provided stable convergence and optimal mAP performance across all defect types.

Reviewer#2, Concern # 12: On page 12, augmentation includes rotations of −45 to 45 degrees. Were other augmentations tried (e.g., brightness changes, blur, cutout)?

Author response: We appreciate the reviewe

---

## [Editor Report · Decision Letter 1]

13 Aug 2025

Vision Transformer and Mamba-Attention Fusion for High-Precision PCB Defect Detection

PONE-D-25-26195R1

Dear Dr. Choi,

We’re pleased to inform you that your manuscript has been judged scientifically suitable for publication and will be formally accepted for publication once it meets all outstanding technical requirements.

Kind regards,

Azim Uddin, Ph.D.

Academic Editor

PLOS ONE
---

## [Editor Report · Acceptance letter]

PONE-D-25-26195R1

PLOS ONE

Dear Dr. Choi,

I'm pleased to inform you that your manuscript has been deemed suitable for publication in PLOS ONE. Congratulations! Your manuscript is now being handed over to our production team.

Kind regards,

on behalf of

Dr. Azim Uddin

Academic Editor

PLOS ONE